# Characterization of Agarose Gels in Solvent and Non-Solvent Media

**DOI:** 10.3390/polym15092162

**Published:** 2023-04-30

**Authors:** Denis C. D. Roux, Isabelle Jeacomine, Guillaume Maîtrejean, François Caton, Marguerite Rinaudo

**Affiliations:** 1Univ. Grenoble Alpes, CNRS, Grenoble INP (Institute of Engineering Univ. Grenoble Alpes), LRP, 38000 Grenoble, Franceguillaume.maitrejean@univ-grenoble-alpes.fr (G.M.);; 2NMR Centers of RMN-ICMG (FR2607), CERMAV-CNRS, BP53, 38000 Grenoble, France; isabelle.jeacomine@cermav.cnrs.fr; 3Biomaterials Applications, 38000 Grenoble, France

**Keywords:** agarose, methyl substituent, DSC, NMR, water retention, rheology, gelation hysteresis, gel in ethanol

## Abstract

Agarose is known to form a homogeneous thermoreversible gel in an aqueous medium over a critical polymer concentration. The solid-liquid phase transitions are thermoreversible but depend on the molecular structure of the agarose sample tested. The literature has mentioned that agarose gels could remain stable in non-solvents such as acetone or ethanol. However, there has been no characterization of their behavior nor a comparison with the gels formed in a good solvent such as water. In the first step of this article, the structure was characterized using ^1^H and ^13^C NMR in both D_2_O and DMSO-*d6* solvents. DMSO is a solvent that dissolves agarose regardless of the temperature. First, we have determined a low yield of methyl substitution on the D-galactose unit. Then, the evolution of the ^1^H NMR spectrum was monitored as a function of temperature during both increasing and decreasing temperature processes, ranging from 25 to 80 °C. A large thermal hysteresis was obtained and discussed, which aided in the interpretation of rheological behavior. The hysteresis of NMR signals is related to the mobility of the agarose chains, which follows the sol/gel transition depending on the chains’ association with H-bonds between water and the -OH groups of agarose for tightly bound water and agarose/agarose in chain packing. In the second step of the study, the water in the agarose gel was exchanged with ethanol, which is a non-solvent for agarose. The resulting gel was stable, and its properties were characterized using rheology and compared to its behavior in aqueous media. The bound water molecules that act as plasticizers were likely removed during the exchange process, resulting in a stronger and more brittle gel in ethanol, with higher thermal stability compared to the aqueous gel. It is the first time that such gel is characterized without phase transition when passing from a good solvent to a non-solvent. This extends the domains of application of agarose.

## 1. Introduction

Agarose is a natural polysaccharide extracted from red seaweed and is known to form a gel in aqueous media. Agarose has found extensive use in diverse fields, including electrophoresis (primarily for DNA), as well as in food engineering, tissue engineering and biotechnology [1,2,3]. It can be solubilized in water or D_2_O (for NMR study) at high temperature or melting temperature (T_m_~80–90 °C) and forms a strong, homogeneous gel upon cooling below the gelling temperature (Tg), provided that the concentration is above a critical concentration (C_0_) which is larger than the overlap concentration (C*) [4,5]. The stabilization of the gel is achieved through a hydrogen-bond network involving -OH groups in an associated double helical structure and water-agarose-OH groups. The reversible process of sol-gel transition exhibits a significant hysteresis, which can be attributed to the coil-helix conformation and subsequent aggregation-packing of double helices. This phenomenon has been previously observed with κ-carrageenans [6,7,8]. The physical characteristics of agarose gel, such as T_g_ and T_m_, depend on the chemical structure of agarose chains, which may contain methyl, acetyl, sulfate, or pyruvate substituents. The agarose gel can be analyzed by NMR in its coiled conformation when it is soluble in DMSO [9,10,11,12]. Agarose, on the other hand, is not soluble in ethanol generally regarded as a non-solvent and a precipitant [13].

Agarose is a linear polysaccharide made of repeating units of agarobiose that is extracted from boiled red algae.
-->3)-β-D-galactose-(1-->4)-3,6-anhydro-α-L-galactose-(1-->

The changes observed in the NMR signals with increasing temperature are linked to the formation and disruption of intra- and inter-molecular H-bonds between agarose chains. These changes are also influenced by the role of water in agarose gel formation, as indicated by relaxation experiments [14,15,16,17,18,19]. It exhibits a sol-gel transition in good solvents such as water or D_2_O at a specific concentration, influenced by factors such as temperature and salt concentration. The physical properties of water or D_2_O gels are well known, but there is a lack of knowledge regarding the comparative behavior and properties of agarose gels formed in non-solvents such as ethanol. Previous literature only briefly discussed the stability of agarose gels in non-solvents such as acetone and ethanol [20].

This study aims to investigate the behavior and properties of agarose gels formed in solvents and non-solvents using rheology. To determine the chemical structure of the agarose used in this study and to identify any possible substituents on the main chain, NMR spectroscopy is utilized. The formation of agarose gels is studied in both water and ethanol, and NMR spectrometry enables the determination of the chemical structure of agarose. Additionally, in order to supplement the assignments provided in the literature, some of the various ^1^H and ^13^C signals were assigned, particularly in D_2_O [7,9,10,13].

## 2. Materials and Methods

### 2.1. Agarose Samples Preparation

Agarose was provided by Sigma Chemicals (Agarose HGTP, N° A-3893/TYPEVI), having a high melting temperature. It was used without purification and solubilized in D_2_O for NMR or deionized water for rheology after heating for 1 h at 90 °C. For agarose characterization, agarose was also solubilized directly in DMSO-*d6* for NMR measurements.

For rheology, the solution was poured into a cylindrical mold, and the temperature was lowered to 15 °C in order to obtain a stable gel with a cylindrical shape adapted for measurements. Samples were then stored at ambient temperature and used within 24 h. In the conditions tested (especially the agarose concentration at 10 mg/mL), no syneresis was observed. For rheology, water exchange with ethanol was performed by immersing the small cylindrical sample in ethanol for 1 week, and the medium was exchanged daily.

For NMR, the solution in D_2_O is introduced in the NMR tube at high temperatures. The exchange of D_2_O against ethanol is performed directly in the NMR tube, covering the gel formed at low temperatures with ethanol for 1 week with the daily exchange of ethanol.

### 2.2. NMR Experiments

The sample was dissolved in D_2_O (5 mg/mL) and in dimethylsulfoxide-*d6* (7.6 mg in 0.6 mL) for NMR characterization. The spectra were obtained at 80 °C and analyzed with chemical shifts assignment for ^13^C and ^1^H. The temperature evolution of spectra in D_2_O was studied to compare with the hysteresis established by rheology. After stabilization at low temperature, a temperature increase was imposed from 25 °C up to 80 °C at a rate of 0.5 °C/min. After 30 min at a high temperature, the temperature was decreased to 25 °C at the same rate. For this experiment, the agarose concentration in D_2_O is 10 mg/mL. In such conditions, the evolution of ^1^H mobility was studied in relation to the sol-gel transition in D_2_O. To quantify this evolution, a pre-determined amount of dimethylsulfoxide (DMSO) was added to enable the calibration of the signal amplitudes of the agarose protons.

The DEPTQ [21] and proton NMR spectra were recorded with a Bruker Avance 400 spectrometer operating at a frequency of 100.618 MHz for ^13^C and 400.13 MHz for ^1^H. The solvent residual peaks of HOD and DMSO-*d_6_* were used as internal standards at 4.8 ppm at 298 K and 4.25 ppm at 353 K for D_2_O, and at 2.5 ppm for DMSO-*d_6_* for ^1^H and 39.51 ppm for DMSO-*d_6_* for ^13^C NMR.

The proton spectra were recorded with a 4000 Hz spectral width, 65,536 data points, 8.19 s acquisition times, a 1 s relaxation delay and 16 scans. DEPTQ spectra were recorded using 90-degree pulses, 20,161 Hz spectral width, 65,536 data points, 1.62 s acquisition time, a 2 s relaxation delay and 6000 scans.

The ^1^H and ^13^C-NMR assignments were based on ^1^H-^1^H homo-nuclear and ^1^H-^13^C hetero-nuclear correlation experiments (correlation spectroscopy, COSY; hetero-nuclear multiple-bond correlation, HMBC; hetero-nuclear single quantum correlation, HSQC). They were performed with a 4000 Hz spectral width, 2048 data points, 0.255 s acquisition time, and a 1–1.5 s relaxation delay; 32–196 scans were accumulated.

### 2.3. Water Regain and Swelling Degree

The degree of swelling was determined from the weight of swollen gel (Wh) in the solvent considered (H_2_O or ethanol) and the dried weight (Ws) expressed in mL solvent/g dried gel taking into account the ethanol density (ethanol d = 0.79). The dried weight was obtained after 2 h at 120 °C.

### 2.4. Differential Scanning Calorimetry

To complete the analysis of the regained water, the amount of frozen water was evaluated by Differential Scanning Calorimetry (DSC) using a Mettler Toledo DSC 821 [22,23]. All experiments were carried out using the following protocol: (1) cooling the sample from 25 °C to −50 °C at −1 °C/min, (2) isothermal at −50 °C for 10 min and (3) heating from −50 °C to 25 °C at 1 °C/min. A nitrogen flow atmosphere was imposed at 60 mL/min to maintain a stable temperature and avoid fluctuations. After calibration with deionized water, it was shown that no residual water was left in the sample after ethanol exchange.

### 2.5. Rheology

Rheology was performed on an ARES-G2 rotational rheometer (TA Instruments, New Castle, DE, USA) with plates geometries of 25 mm diameters in 2 configurations: 1 using the Advanced Peltier System, APS (−10 °C to 150 °C) and the other using a Forced Convection Oven, FCO (−150–600 °C).

During temperature increases, it is important to prevent solvent evaporation. The plate systems used, as illustrated in Figure 1, were equipped with a cup at the bottom where Paragon S3 silicon oil (Paragon Scientific Ltd., Birkenhead, UK) was poured to cover the gel and prevent water evaporation. The silicon oil viscosity used to prevent water evaporation changed from 3.7 mPa·s at 20 °C to 1.2 mPa·s at 80 °C. No difference was observed in the elastic and viscous moduli of the gels measured at ambient temperature with and without silicon oil. Furthermore, no slip was observed on the sheared samples, thanks to the ribbed geometry. The temperature ramps applied to the samples were identical to that applied to the RMN ones, with a polymer concentration of 10 g/L and an imposed ramp rate of 0.5 °C/min, in order to reduce the effect of thermal tool inertia.

The experiments were conducted, paying particular attention to the contact between the gel and the plates by applying a small axial force between 0–0.1 N perpendicular to the plate surface. Temperature ramps were applied differently depending on the geometry used. For the smooth plate-plate geometry (Figure 1a), the temperature ramps were controlled by the APS system, while for the radial ribbed plates, temperature ramps were controlled by the FCO system (Figure 1b). No significant difference was observed between the two systems. The water and ethanol gels exhibited different behaviors during the temperature ramps. During the temperature ramps, the water and ethanol gels behaved differently. The water gels did not require any special attention during the experiments, but the ethanol gel showed a drastic change in sample size as the temperature increased due to solvent evaporation. This made rheometric measurements for the ethanol gel difficult as the temperature increased. Due to this difficulty in controlling the integrity of the ethanol gel, the present study was limited to 50 °C for the ethanol gel.

## 3. Results and Discussion

### 3.1. NMR Study

Analysis of the agarose sample is performed using 1D, 2D ^1^H and ^13^C NMR spectroscopies. Signal assignments are given in Table 1 and Table 2 for the solvents D-O and DMSO-*d6*. In the tables, G is used for the anhydrogalactose unit, G′ for the D-galactose unit and G″ for the substituted D-galactose unit (Figure 2).

#### 3.1.1. Assignement of ^13^C and ^1^H in D_2_O

The complete assignment for protons and carbons is given in Figure 3 and Figure 4. The DEPTQ permits us to identify carbon when it is engaged in -CH_2_ groups (Figure 4). The attribution of the signals was permitted using other NMR techniques shown in the following. The use of COSY (Figure 5) allows us to assign protons and HSQC to attribute corresponding carbons (Figure 6). The chemical shifts are summarized in Table 1.

#### 3.1.2. Assignement of ^13^C and ^1^H in DMSO-d6

In DMSO-*d6*, with agarose in the coil conformation, the obtained spectrum shows the proton and carbon signals but also allows us to identify the -OH groups (Figure 7 and Figure 8). The identification of ^1^H and ^13^C signals is summarized in Table 2.

The assignments given in Table 2 are in good agreement with data from the literature [9,11,13,17]. 2D HMBC spectrum was used to locate the methyl substituent (Figure 9). The methyl group is identified on the D-galactose unit, which shifts C-3 to C-6 signals (D-galactose G″ in spectra) and the two H-6 (Table 2). This substitution is discussed in the following paragraph. 

#### 3.1.3. Degree of Substitution on Agarose

Considering the long-distance correlation of ^1^H-^13^C performed in DMSO-*d6*, it is shown that the proton with a thin signal at 3.3 ppm is assigned to a methyl substituent correlated with the C-6 of the D-galactose unit. In the same way, the C from the methyl group correlates with the proton H-6 of D-galactose. Taking into account the integral of this signal in reference to the H-1 of anhydrogalactose, it comes to DS = 0.24 ± 0.01, indicating the degree of substitution of D-galactose units.

#### 3.1.4. Thermal Hysteresis

NMR is an interesting technique to investigate the local mobility of molecules. The spectra given in Figure 10 and Figure 11 show the evolution of the signal amplitudes as well as the chemical shift of HOD (from agarose -OH exchanged to -OD in D_2_O) used as a reference for temperatures from 25 to 80 °C and from 80 to 25 °C.

The H-1 from anhydrogalactose at 5.16 ppm and the methyl group on galactose at 3.3 ppm were selected to draw the temperature dependence. The hysteresis obtained may be discussed in relation to the rheological data (Figure 12). For this experiment, a small defined amount of DMSO is added to allow the comparison of the amplitude of the two protons’ selected signals.

The integrals of the signals corresponding to H-1 of anhydrogalactose unit (or -CH_3_ substituent on galactose unit, not shown) are plotted as a function of temperature in Figure 12 reflecting the modification of chains mobility with a large hysteresis: it shows a transition over 60 °C on increasing temperature and another one under 50 °C on decreasing temperature.

Papers concerning NMR of agarose in aqueous media are based mainly on ^1^H and ^2^H magnetic relaxation studies (T_1_ and T_2_), allowing us to deduce the bound and free water [15,16] without an exact estimation of the fraction of bound water molecules. Interactions are related to the existence of H-bonds in the gel state or double helices aggregates involving agarose-agarose and water-agarose interactions. It was proposed that the large magnetic relaxation dispersion is due to internal water molecules located in the central cavity of the agarose double helix, which stabilize the conformation [14]. Slower self-diffusion of water was also related to the obstructive effect of the agarose network [19]. The dependence of NMR spectra on temperature was mainly investigated in a few papers in relation to gelation [7,17].

Our data clearly show a hysteresis between heating and cooling that is related to rheology, as discussed later. In addition, a small chemical shift (0.023 ppm) is obtained around 65 °C on heating steps attributed to conformational changes, i.e., probably, helix-coil transitions associated with aggregates’ progressive dissociation. The signal related to HOD is also larger in the temperature range up to 65 °C, probably due to gel presence, as suggested previously [19]. This transition was also described by Chavez et al. at around 55 °C and mentioned at 55 °C by Ed-Daoui et al. [14,24]. Our data taken at high temperatures are interpreted as a consequence of the high mobility of the coiled chains at their melting temperature T_m_ around 80 °C. On cooling, the chemical shift located around 70 °C is smaller, interpreted as the helix formation, with a smaller chemical shift of 0.011 and a thin signal for HOD down to 45 °C. From this temperature, aggregates formed in the liquid phase from 65 °C are cross-linked down to gelling temperature around T_g_ = 20 °C. It is known that H-bond-type interactions are formed in D_2_O and that there are stronger than in H_2_O. Then, a double-helical conformation is more stable in D_2_O as obtained for κ-carrageenan [25]. This may justify the small increase in the characteristic temperatures given by NMR compared to rheology.

#### 3.1.5. Thermal Behaviour of Gel in Ethanol

From the influence of increasing temperature on the NMR spectrum, it is shown that the large signal around 5.2 ppm decreases progressively up to 65 °C, followed by the appearance of a thinner signal observed over 65 °C (Figure 13). It seems those signals appearing are related to H-1 of the anhydrogalactose unit based on the chemical shifts obtained in D_2_O. This indicates that only a small fraction of agarose is mobilized when temperature increases over 65 °C (if the signals are compared with those obtained in D_2_O in the same conditions). It is probably related to the release of a few H-bonds in the loose junction zones connecting the stiff aggregates. In these spectra, the HOD signal is related to the exchange of agarose -OH due to the addition of a small amount of D_2_O necessary to lock the sample and the conditions to perform the experiment. This signal remains large, indicating a cross-linked system.

In a separate experiment, it was shown that the gel never melts up to 80 °C when immersed in ethanol. This indicates the absence of H_2_O or D_2_O inserted inside the cavity of double helices or between them; the cooperative H-bonds are more stable, as confirmed by the rheological experiment in the following. Additionally, our observations indicate that the exchange between water and ethanol in the porous gel is reversible, allowing for its use in various water-ethanol mixtures.

### 3.2. Degree of Solvation

The first step was to determine the degree of swelling of the gel in water and, after the exchange, in the presence of ethanol. In water, (Wh − Ws)/Ws= 101 g/g dried gel or 101 mL/g dried gel in agreement with the weight concentration of the solution prepared at 10 g/L. The ethanol content is given by (Wh − Ws) × 0.79/Ws = 95 mL/g dried gel. On the same sample, DSC showed that no water molecules remained in the ethanol gel. It is concluded that the degree of swelling (and consequently the porosity) is nearly the same in both solvent conditions and confirms our previous results [20], even if ethanol is a non-solvent of agarose. Hence during the process, ethanol replaces water inducing very little shrinkage. This clearly indicates that there is no gel collapse as described on chemically cross-linked polymers by Tanaka [26,27].

### 3.3. Rheology of Agarose Gels

#### 3.3.1. Comparison of Gels in Water and Ethanol

Gels stabilized in D_2_O and ethanol were tested as a function of temperature in rheological experiments. Figure 14 shows elastic and viscous moduli as a function of the frequency of formed gels in water (blue) and in ethanol (green) for a concentration of 10 mg/mL at T = 20 °C. Elastic moduli are well above viscous moduli, with a ratio of around 10 and independent of the investigated frequencies, as expected for stiff gels. Ethanol gel is 10 times stronger than water gel even if the swelling volume change in the non-solvent is nearly the same as in water. The characteristic strength of the ethanol gel remains even when the temperature is increased to 50 °C (Figure 15). The moduli at the highest temperature are not shown due to a repeatable experimental problem (see material and methods).

To our knowledge, this is the first time that an ethanol gel has been studied. In fact, few experiments are described in the literature about agarose behavior in the presence of ethanol. For a diluted agarose solution in water containing 6% ethanol, the mesoscopic aggregates observed are more compact in morphology due to the modification of the balance between H-bonds and hydrophobic interactions [5]. In another way, ethanol is used for the coagulation of agarose/DMSO for fibers production [12], and an isoelectric focusing technique was applied for the characterization of gliadins [28] in 45% ethanol for a better resolution. The authors assert that this composition is the highest possible ethanol concentration that does not cause the precipitation of agarose.

To complete the gel study, Figure 16 shows the loss and viscous moduli as a function of the applied strain for a constant frequency of f = 1 Hz. From the amplitude sweep, a weaker linear regime was observed for the ethanol gel (γ~1%) than for the water gel (γ~3%). For both gels, a reduction in moduli at high amplitudes of oscillation is present but with a difference in the loss modulus. Indeed, there is an overshoot in the loss modulus for the ethanol gel but not for the water gel. We interpret this difference as a modification of the structural organization of the water gel and a destruction of the structure of the ethanol gel. This is confirmed by the subsequent experiments done on the ethanol gel, where the viscous and the elastic moduli dropped by a factor of 10 and where the oscillatory stress and strain curves no longer resembled sinusoidal curves, even in the linear regime of the gel identified initially in Figure 16. This interpretation is in agreement with the literature [29,30,31], and it is in favor of the plasticizing role of water in agarose gel. Due to the absence of water in the ethanol gel, it behaves like a brittle gel.

As observed in all measurements of G′ and G″, regardless of temperature, G′ is higher than G″ at the observed frequencies, which is a consequence of the behavior of an elasto-viscous fluid in a semi-diluted regime of high molar mass polymers.

#### 3.3.2. Hysteresis for Sol-Gel Transition in Water

In Figure 17, elastic and loss moduli are tracked between 15 °C to 80 °C at 0.5 °C/min for the water gel. In that figure, three temperature ramps are shown. First increasing the ramp after loading at 20 °C, (1) the sample was brought to 20 °C before being measured in an increasing ramp where the moduli of elasticity and loss slowly decreased before starting to drop at around 60 °C, showing a solid-sol transition at about 78 °C, attributed to Tm. These values confirmed those obtained in NMR. During this ramp, G′~3 × 10^3^ Pa and G″~10^2^ Pa confirmed the ratio of G′/G″ as being 10. (2) When first decreasing the ramp, G′ was constant until about 62 °C (probably at the coil-helix transition), where it increased exponentially with temperature before reaching a constant value of G′ = 8 × 10^3^ Pa, which was higher than the initial value at the first load. At T~45 °C, there was a change of concavity corresponding to the abrupt fall in NMR shown in Figure 12. This behavior was interpreted as being a consequence of the beginning of the cross-linkage of helix aggregates accompanied by the formation of a non-continuous bulk gel. Interestingly, G″ did not change too much until this same temperature of 45 °C, where it starts to increase during the decreasing temperature ramp. This behavior was interpreted as the starting point of the gel continuum formation. At T~20 °C, corresponding to Tg, G″~2 × 10^2^ Pa, which was also a higher value than at the initial load. (3) For this last increasing ramp, G′ and G″ showed a similar behavior as in the first ramp but with higher values confirming the strongest strengthening of the gel at the low-temperature rate. This increase in G′ at low temperatures was due to the temperature history of the sample firstly being prepared without any control of the temperature. The physical properties of the gels at lower temperatures are, in fact, dependent on the kinetic of gelation playing on the degree of double-helix packing and that of junction zone formation.

## 4. Conclusions

Agarose is a well-known polysaccharide extracted from red algae. After alkaline treatment, it becomes soluble in aqueous media at high temperatures (over a melting temperature T_m_ characterizing the gel-sol transition). Then, it forms strong physical gels at low temperatures (lower than T_g_, corresponding to the sol-gel transition). Agarose is mainly used in biotechnologies taking advantage of the large rigid pores in the gel state.

Nevertheless, the physical properties of agarose are directly related to their chemical structure and especially to the presence of substituents on the main chains. In the first part of this paper, the chemical structure of agarose was perfectly determined by ^1^H and ^13^C NMR spectroscopies. The presence of a methyl substituent was determined on the D-galactose repeat unit with a degree of substitution of 0.24.

Secondly, the influence of temperature on ^1^H NMR spectra during heating and cooling was investigated, allowing us to draw a hysteresis able to help in the analysis of rheological behavior. From those results, it was suggested that on heating, the helix-coil transition occurs around 60 °C followed by melting at T_m_~80 °C. This helix-coil transition was also associated with a chemical shift of the selected H-1 signal of the anhydrogalactose unit but also of the other protons. On cooling, a smaller chemical shift was observed, but the integral H-1 signal indicated a slow decrease to around 50 °C, corresponding to the progressive formation of double helix aggregates. Then, the rapid decrease of the signal to the gelling temperature T_g_~25 °C corresponded to the mobility decrease during the network formation through the progressive cross-linkage of aggregates. It is necessary to mention that there was a large dynamic process of cross-linkage between the double helix aggregates, which progressively modified the packing of aggregates and the gel properties at low temperatures and was also related to syneresis.

In the next step, the homogeneous gel formed at the tested concentration (10 g/L in water) was characterized by its degree of swelling, and then the water was exchanged with ethanol, a non-solvent of agarose. This original gel had nearly the same porosity as the gel formed in water. This will allow us to extend the domains of application of agarose as a gelling polymer and porous material. The NMR indicates that this gel did not melt until 80 °C (limit in ethanol), as confirmed in a separate experiment. This means that there was no collapse in ethanol, as previously mentioned [20]. Furthermore, it has been shown that water-ethanol exchange in agarose gels is reversible.

The last part of this study concerns the rheology of agarose gel forms in water and ethanol. At ambient temperature, the ethanol gel is stronger but clearly more brittle than the gel in water identified in oscillatory strain sweep experiments. This result is related to the plasticizing effect of water in aqueous media, which is suppressed after ethanol exchange. This confirms that the aggregates of double helices and network junctions are stiffer and thermally stable.

Then, the influence of temperature on the G′ and G″ moduli in the linear regime is studied, particularly on gels formed in water. In the increasing ramps, water gel shows a clear melting temperature T_m_ = 78 °C in agreement with NMR. During the decreasing ramp, the G′ and G″ behaviors are correlated with two temperatures, T = 62 °C and T = 45 °C, identified by NMR and interpreted respectively as the beginning of the coil-to-helix transition followed by aggregate formation. In this domain of temperature, G″ remains nearly constant. But from 45 °C and when the temperature is decreased, a bulk gel is progressively formed due to cross-linkage of the aggregates. Therefore, the combination of NMR and rheology techniques proves to be complementary and beneficial in identifying the various stages of the reversible sol-gel transition of agarose as it relates to temperature.

In conclusion, this study investigates the mechanism of the reversible sol-gel transition on a well-characterized agarose sample through NMR and rheology techniques. The combination of these two methods yields a strong correlation for describing the gelation and melting processes. To extend the study of the phase diagram, it is recommended to apply these techniques to varying concentrations of agarose. Furthermore, a new stable gel in ethanol has been discovered, allowing for the use of agarose as a porous material in non-solvent media. Through the properties of the ethanol gel, the influence of water molecules in the formation of double helix and aggregates in agarose-agarose interactions is clearly identified.

## Figures and Tables

**Figure 1 polymers-15-02162-f001:**
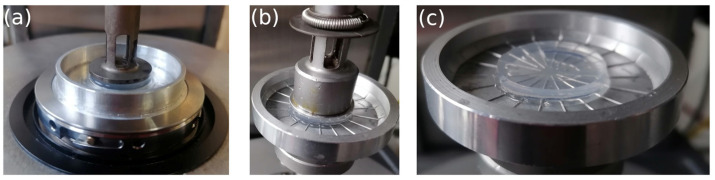
Photographs depicting the two different plate geometries utilized for gel rheology. The image labeled (**a**) on the left displays the smooth plate-plate, which is mounted onto the Peltier system control. On the right, the photo labeled (**b**) illustrates the radial ribbed plates that are thermally controlled by the FCO controller of the ARES G2. Both geometries have a diameter of 25 mm and consist of a cup on the bottom part and a plate on the upper part, between which the gel is placed before being covered by a layer of silicon oil to prevent water evaporation. The gel’s position on the ribbed plate is shown in the photo labeled (**c**) on the right.

**Figure 2 polymers-15-02162-f002:**
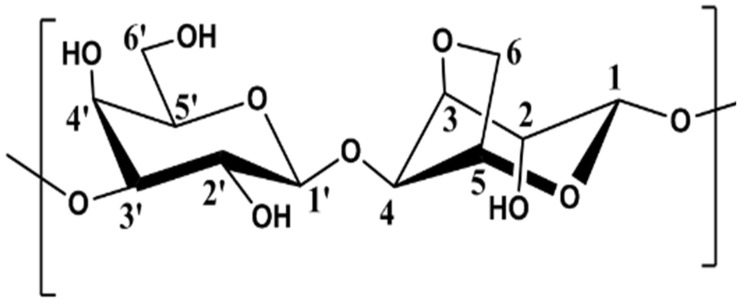
Representation of the repeated unit in agarose.

**Figure 3 polymers-15-02162-f003:**
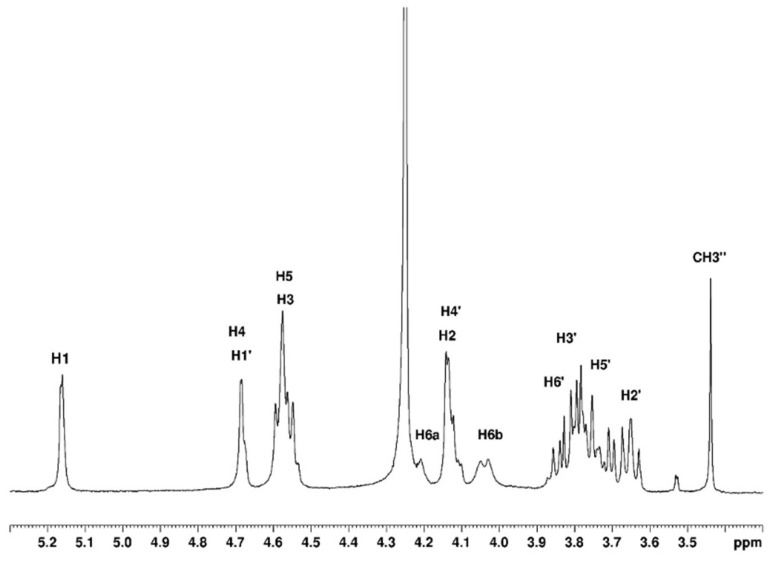
Proton NMR of agarose in D_2_O at 80 °C.

**Figure 4 polymers-15-02162-f004:**
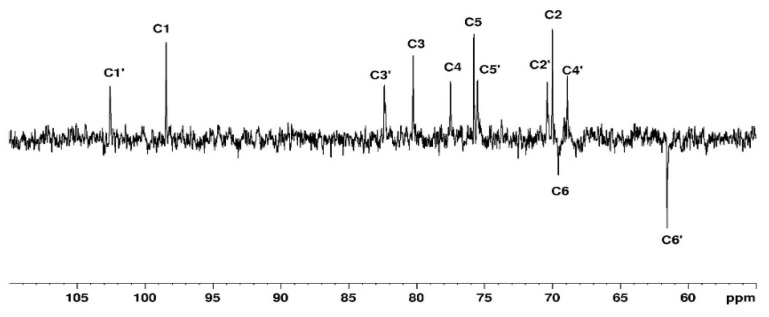
DEPTQ spectrum of agarose in D_2_O at 80 °C.

**Figure 5 polymers-15-02162-f005:**
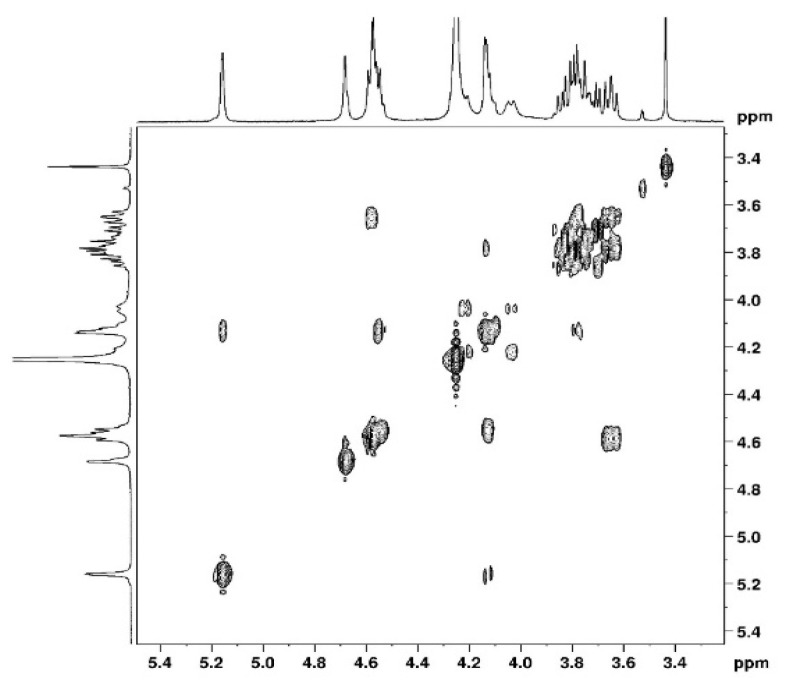
2D COSY NMR spectrum of agarose in D_2_O at 80 °C.

**Figure 6 polymers-15-02162-f006:**
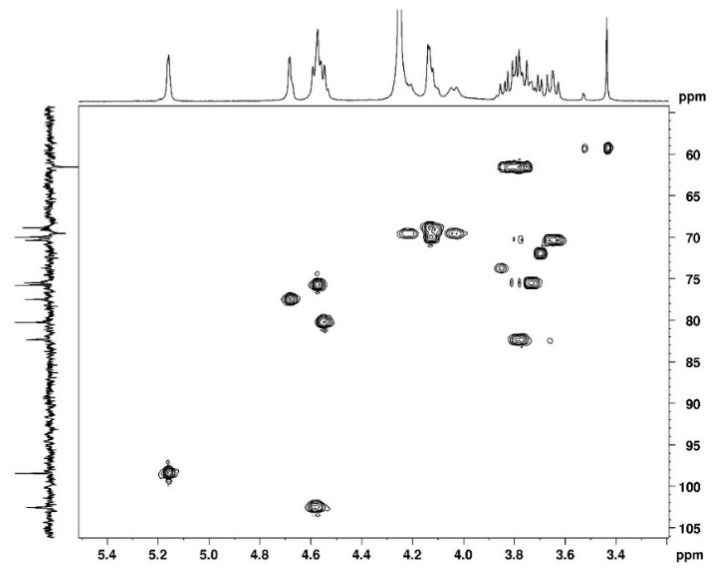
2D HSQC NMR spectrum of agarose in D_2_O at 80 °C.

**Figure 7 polymers-15-02162-f007:**
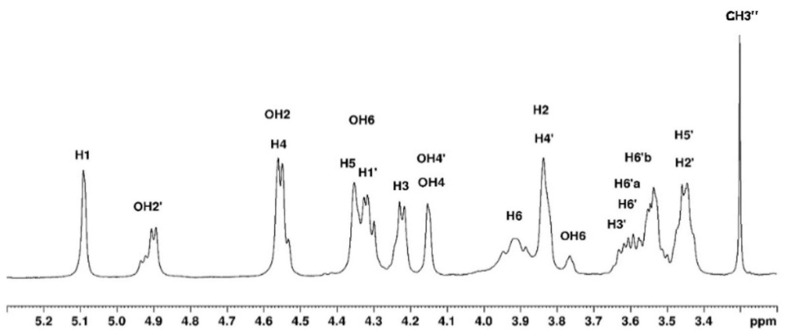
Proton NMR spectrum of agarose in DMSO-*d6* at 80 °C.

**Figure 8 polymers-15-02162-f008:**
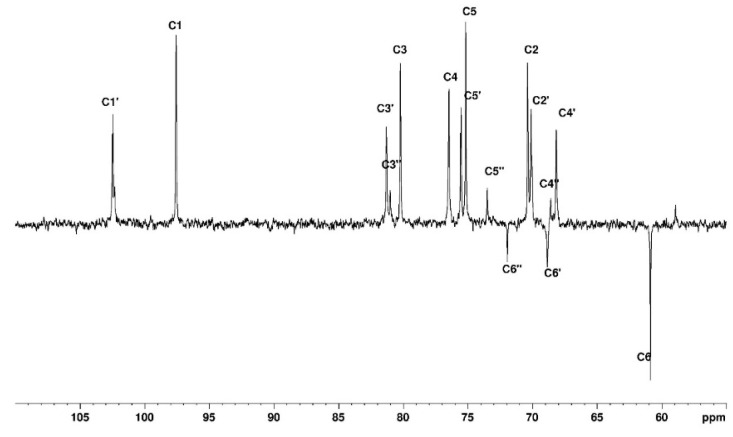
DEPTQ spectrum for agarose in DMSO-*d6* at 80 °C.

**Figure 9 polymers-15-02162-f009:**
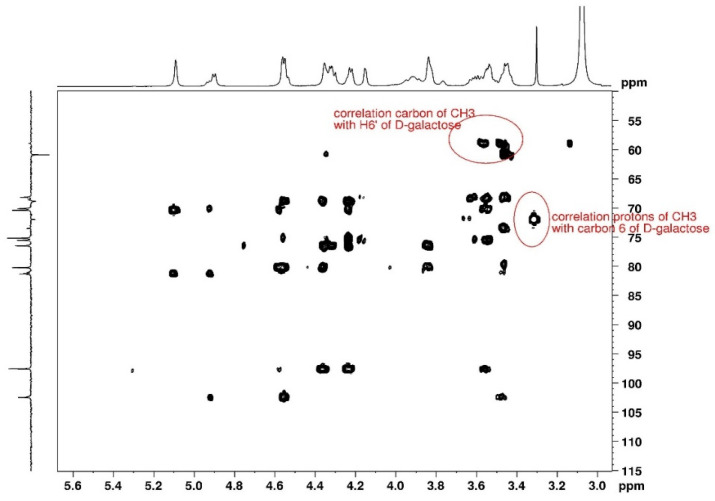
2D HMBC NMR spectrum of agarose in DMSO-*d6* at 80 °C used for location of the methyl substituent.

**Figure 10 polymers-15-02162-f010:**
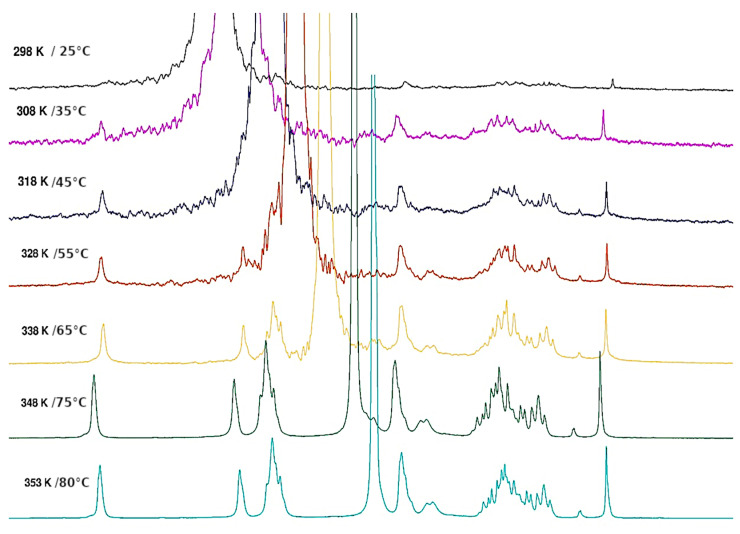
Temperature increase on ^1^H NMR spectrum of agarose in D_2_O from 25 °C to 80 °C.

**Figure 11 polymers-15-02162-f011:**
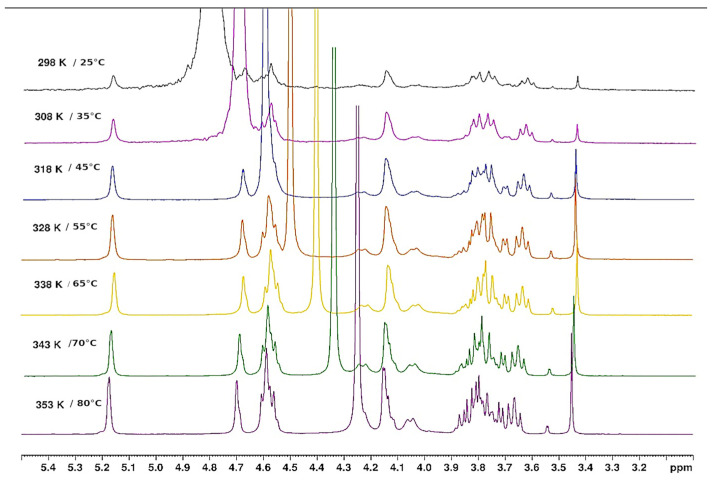
Temperature decreases for the ^1^H NMR spectrum of agarose in D_2_O from 80 °C to 25 °C.

**Figure 12 polymers-15-02162-f012:**
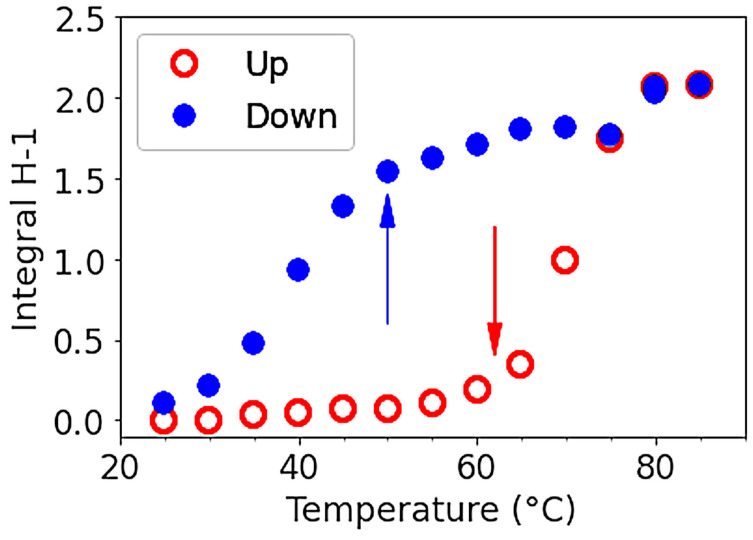
H-1 integrals of anhydrogalactose as a function of temperature on heating and cooling from 25 to 85 °C. The open red circles represent the heating process, while the filled blue circles represent the cooling process.

**Figure 13 polymers-15-02162-f013:**
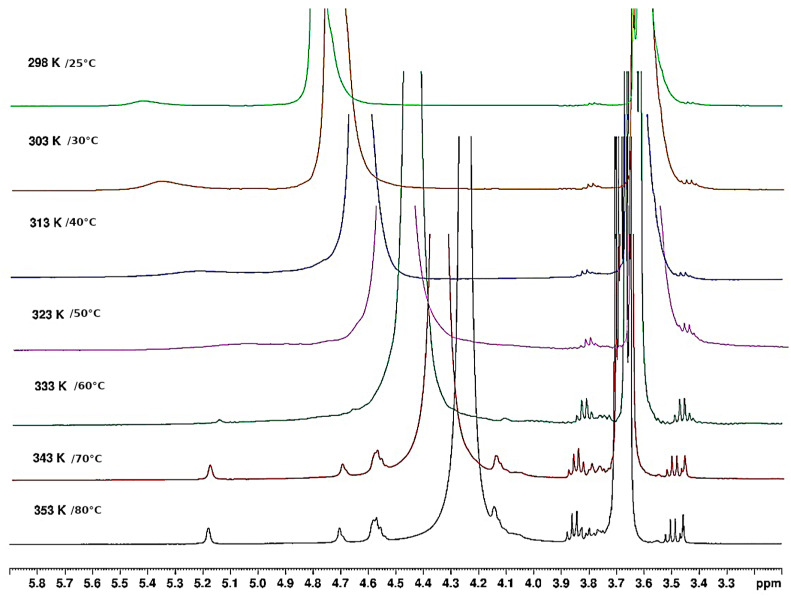
^1^H NMR agarose gel in ethanol. Evolution during temperature increase from 25 °C to 80 °C.

**Figure 14 polymers-15-02162-f014:**
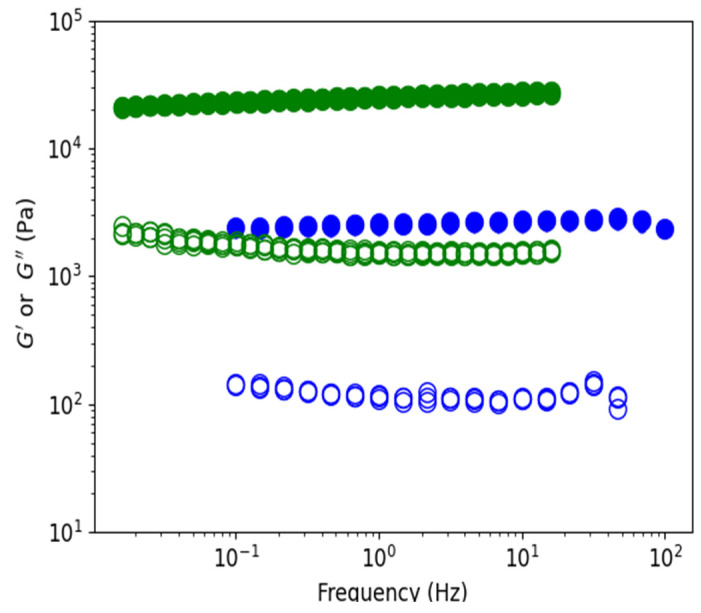
Rheological characteristics G′ (●) and G″ (○) as a function of the frequency of gels formed in water (blue) and in ethanol (green). The concentration was 10 g/L at 20 °C for 0.1% of the applied strain.

**Figure 15 polymers-15-02162-f015:**
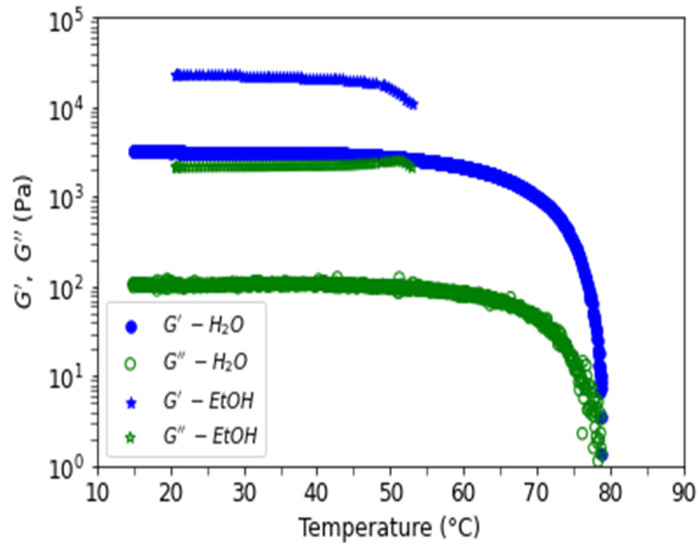
Rheological moduli G′ and G″, at 1 Hz and 0.1% strain, for agarose gels in water (●, ○) and in ethanol (★, ☆) as a function of temperature. The imposed temperature rate ramp is equal to 0.5 °C/min.

**Figure 16 polymers-15-02162-f016:**
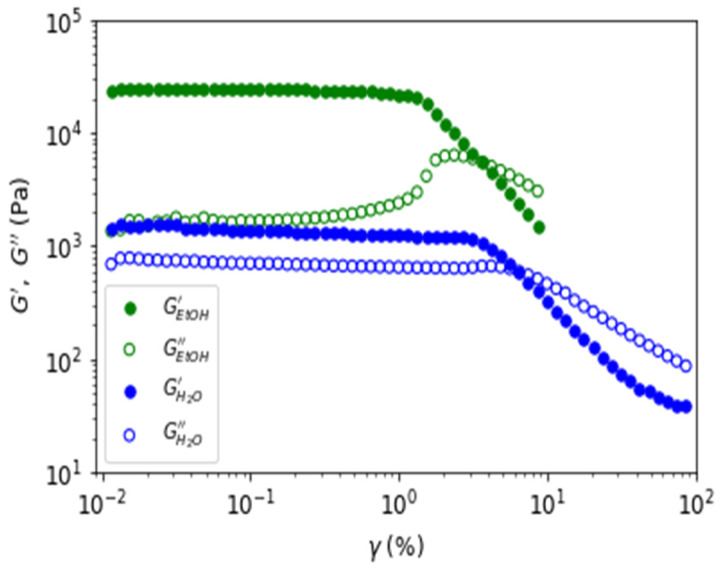
Elastic (full dots) and viscous (open circles) moduli as a function of the strain for a constant frequency of 1 Hz for the water (in blue) et ethanol gels (in green).

**Figure 17 polymers-15-02162-f017:**
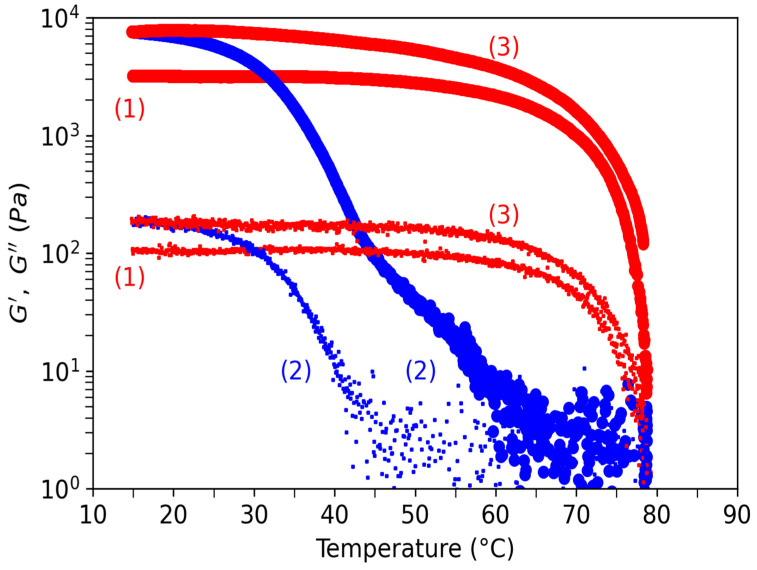
Elastic (large dots) and viscous (small dots) moduli as a function of the applied temperature for the water gel. Increasing temperatures correspond to the red symbols, and decreasing temperatures correspond to the blue symbols. The numbers indicate the time sequence of the applied temperature ramp, (1) first increase, (2) first decrease, (3) second increase.

**Table 1 polymers-15-02162-t001:** Chemical shifts of proton and carbon signals for agarose in D_2_O at 80 °C.

	**C1**	**C2**	**C3**	**C4**	**C5**	**C6**	**CH_3_**
G′	102.63	70.38	82.33	68.93	75.51	61.56	
G″	102.63	70.38	82.33	68.93	75.51	71.89	59.2
G	98.52	70.05	80.31	77.6	75.7	69.64	
	**H1**	**H2**	**H3**	**H4**	**H5**	**H6**	**CH_3_**
G′	4.58	3.65	3.78	4.14	3.73	3.80	
G″	4.58	3.65	3.78	4.14	3.73	3.70	3.43
G	5.16	4.13	4.54	4.68	4.57	4.22 ^a^–4.04 ^b^	

**Table 2 polymers-15-02162-t002:** Chemical shifts of proton and carbon signals for agarose in DMSO-*d6* at 80 °C.

	**C1**	**C2**	**C3**	**C4**	**C5**	**C6**	**CH_3_**
G′	102.5	70.13	81.35	68.11	75.54	60.8	
G″	102.5	70.13	80.99	68.63	73.52	71.96	58.9
G	97.54	70.37	80.23	76.53	75.12	68.9	
	**H1**	**H2**	**H3**	**H4**	**H5**	**H6**	**CH_3_**
G′	4.31	3.45	3.54	3.83	3.44	3.55	
G″	4.31	3.45	3.47	3.76	3.63	3.54 ^a^–3.46 ^b^	3.3
G	5.09	3.83	4.22	4.34	4.35	3.91	

## Data Availability

Rheological data can be obtained from D.Roux (Laboratoire Rhéologie et Procédés) and NMR data from I. Jeacomine (CERMAV-CNRS).

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
