# Peer review of "Characterization of Agarose Gels in Solvent and Non-Solvent Media"

_polymers, 2023, doi:10.3390/polym15092162_

Round 1

Reviewer 1 Report

The authors used NMR and rheology to deeply characterize agarose gel and sol in solvent and non-solvent. The experiments were well designed and results sounded good. However something more needed to be clarified before publish:

1.  Agarose in sol could well be characterized by liquid NMR. The residual dipolar interaction appears in gel state, which is not suitable characterized by liquid NMR. Here in this work, some peek disappeared during gelation process. The observed peak should be indicated in Figure 12. While connected with rheology data, gel network information became more important, which could be better interpreted by solid state NMR.

2. The x-axis of Figure 10 was missing.

3. FCO of ARES G2 was not used in current work, which did not need to mention. Figure in middle of Figure 1 should be described.

Reviewer 2 Report

Agarose is a polysaccharide that can form a gel in aqueous solution above a certain concentration. In this paper, authors characterized the structure of agarose using 1H and 13C NMR at different temperature. By further comparing the properties in water and ethanol, the different stability of agarose in solvent and non-solvent was well explained.

The results found in this paper would be great helpful to enhance the fundamental understanding of agarose, which may contribute to its further application. References are cited appropriately, however, some of them can be updated with recently published papers. The data in this work is mostly collected from NMR and rheology test with well designed parameters. With the conclusion supported by the data, I recommend its publishing. Please see some detailed comments below.

1. For Figure 1, it would help to understand the test process if authors can label the different parts in the figure.

2. Please explain more about the NMR results. Why were Figure 5 and Figure 6 not mentioned in the manuscript?

3. Is it 1H NMR or 1H NMR? Please make it consistent, such as the caption of Figure 10 and Figure 11.

4. The temperature is defined in two different units, please make it consistent.

5. For Figure 17, the color of legend for G and G' is a little confused. What about the brown and blue ones?

6. Please talk more about the significance for this study in the conclusion section.

Minor issues should be addressed, such as some grammar error.
